# High Intensity Drinking (HID) Assessed by Maximum Quantity Consumed Is an Important Pattern Measure Adding Predictive Value in Higher and Lower Income Societies for Modeling Alcohol-Related Problems

**DOI:** 10.3390/ijerph20043748

**Published:** 2023-02-20

**Authors:** Thomas K. Greenfield, Camillia K. Lui, Won K. Cook, Katherine J. Karriker-Jaffe, Libo Li, Sharon C. Wilsnack, Kim Bloomfield, Robin Room, Anne-Marie Laslett, Jason Bond, Rachael Korcha

**Affiliations:** 1Alcohol Research Group, Public Health Institute (PHI), 6001 Shellmound St., Suite 450, Emeryville, CA 94608, USA; 2Community Health & Implementation Research Program, Research Triangle Institute, Berkeley Office, CA 94704, USA; 3Department of Psychiatry and Behavioral Science, School of Medicine and Health Sciences, University of North Dakota, Grand Forks, ND 94704, USA; 4Centre for Alcohol and Drug Research, School of Business and Social Sciences, Aarhus University, 2400 Copenhagen, Denmark; 5Centre for Alcohol Policy Research, School of Psychology and Public Health, La Trobe University (Melbourne Campus), Bundoora, VIC 3086, Australia; 6Centre for Social Research on Alcohol and Drugs, Department of Public Health Sciences, Stockholm University, 106 91 Stockholm, Sweden

**Keywords:** alcohol, drinking patterns, alcohol problems, cross-national, surveys, measurement, policy

## Abstract

Adjusting for demographics and standard drinking measures, High Intensity Drinking (HID), indexed by the maximum quantity consumed in a single day in the past 12 months, may be valuable in predicting alcohol dependence other harms across high and low income societies. The data consisted of 17 surveys of adult (15,460 current drinkers; 71% of total surveyed) in Europe (3), the Americas (8), Africa (2), and Asia/Australia (4). Gender-disaggregated country analyses used Poison regression to investigate whether HID (8–11, 12–23, 24+ drinks) was incrementally influential, beyond log drinking volume and HED (Heavy Episodic Drinking, or 5+ days), in predicting drinking problems, adjusting for age and marital status. In adjusted models predicting AUDIT-5 for men, adding HID improved the overall model fit for 11 of 15 countries. For women, 12 of 14 countries with available data showed an improved fit with HID included. The results for the five Life-Area Harms were similar for men. Considering the results by gender, each country showing improvements in model fit by adding HID had larger values of the average difference between high intensity and usual consumption, implying variations in amounts consumed on any given day. The amount consumed/day often greatly exceeded HED levels. In many societies of varying income levels, as hypothesized, HID provided important added information on drinking patterns for predicting harms, beyond the standard volume and binging indicators.

## 1. Introduction

Improving alcohol intake pattern measurements is vital for epidemiological studies [1,2,3,4] including those in both developing and developed countries [5]. Accurate assessment of alcohol intake and heavy drinking is critical in efforts to better understand drinking outcomes [6]. Self-reported alcohol use accounts for only 40–60% of alcohol sales [7] and more recently, depending on the measures used, coverage has been found as low as 22 to 34% [8,9,10]. De Vries, Lemmens, Pietinen, and Kok [11] argued that the under-reporting of heavy drinking likely contributes considerably to overall under-reporting. Episodic and occasional heavy drinking may be missed by measures that only assess the usual number of drinks per drinking day or average volume of alcohol consumption [9,12]. Indeed, volume of reported alcohol intake increases when measures include atypical heavy as well as typical consumption [13,14,15,16] and measures including HID levels better predict alcohol related problems than either usual quantity–frequency or weekly diary measures [16]. Capturing the variability or pattern of alcohol use is important and can be aided by assessing amounts well beyond HED thresholds [1,17]. This is because those with High Intensity Drinking (HID) occasions, as captured by the maximum quantity consumed in any day of a given period, have higher variability of amounts consumed on any day over that period.

There has recently been an upwelling of attention to High Intensity Drinking, with calls for more HID research [18,19]. A considerable amount of the new literature on HID has remained limited to youth and young adult samples [20,21,22]. However, some HID research in the US has begun to include the general population [23,24]. To our knowledge there has been little multinational research on HID to date. One recent cross-national study, however, has considered the proportion of 8+/6+ (men/women) drinking compared to overall intake [25]. Maximum quantity consumed, as an indicator of HID, has been identified in the US as a valuable pattern measure [26] and together with volume, HID is a predictor of Alcohol Use Disorders (AUD) in the US population [27] and in one cross-national study in populations of Mexican descent on either side of the US border [28]. The maximum quantity consumed, indexing HID, has also been found to be a valuable measure to predict other social and health harms [2] among current drinkers in the general population and other subgroups. Earlier work with the US general population and treated and concerned drinkers found that not only Heavy Episodic Drinking (HED), but higher levels such as 8+, 12+, and even 24+ drinks in a day added to predicting DSM-IV alcohol dependence and abuse at higher volumes of consumption [29]. However, there is a dearth of multinational studies on HID, and especially on its relationship to alcohol-related problems. Based on prior studies limited to the US [27] and its southern border [28], we hypothesized that HID, here indexed by maximum quantity consumed, would significantly augment standard intake and binge (5+) drinking measures, improving the prediction of symptoms of AUD and Life-Area Harms in a wide range of low, middle, and high income countries.

## 2. Materials and Methods

### 2.1. Data Sources

The survey samples were collected under the multi-country GENACIS (Gender, Alcohol and Culture, an International Study) project [5,30,31,32]. The questions were developed first in English, then translated into the main language of the society that was surveyed, followed by back-translation with a check for accuracy and culturally accessible meaning. Guidelines for question translation were adapted from the World Health Organization (WHO) strategies (see the GENACIS website: https://med.und.edu/genacis/ (accessed on 28 December 2022) and Wilsnack et al. [5]). A total of 17 countries collected data on the maximum quantity consumed in a single occasion in the past 12 months. A subset of 14 of these 17 countries also collected measures of life-area Harms and a slightly different subset of 15 countries collected data on the five AUDIT dependence items—see measures. The countries included were also required to have a measure of volume of consumption as well as a measure of HED, sometimes called binge drinking (implemented as frequency of 5 or more (5+) drinks per occasion). Countries (or more accurately, societies, since not all sampling frames were the whole country) with enough data available for use in predicting either the harms or the AUDIT dependence scale included higher- and lower-income countries in six geographic areas (see Table 1), including Africa (Nigeria and Uganda), Australia, Asia (India, Sri Lanka, and Kazakhstan), Europe (Finland, Isle of Man, and Sweden), and the Americas, both South and North, (Argentina, Brazil, Costa Rica, Nicaragua, Uruguay, Mexico, Canada, and the USA). The survey design methods were mostly similar across countries, using a commonly developed instrument, although there was some variation (see Table 1 for details). Surveys in each country were conducted between 1998 and 2007 with appropriate ethics approvals obtained in each country. Many sampling frames were national or nearly national, whereas others represented a state (e.g., in India—Karnataka state) or selected areas within a country. For example, the East Kazakhstan survey likely sampled mostly Russian-language groups. Regional studies generally focused on large population centers within the country. In several cases, the areas within the country account for more than 50% of the country’s total population.

Some surveys were conducted face-to-face by trained interviewers; others involved telephone surveys. In some cases by telephone, the sampling used random digit dialing techniques and in other instances, the sampling was register based. In many cases, multi-stage cluster sampling was used, stratifying by district or some other regional descriptor. In the majority of cases, one individual in the age range (typically over 18, but sometimes with an upper age cap of 65 or 75) was randomly or systematically selected per enumerated or selected household. The analyses that follow used current (i.e., within the past 12 months) drinkers only, with an average sample size across countries of 916 for men and 909 for women. Per the GENACIS study objectives, nearly all datasets, with the exception of Brazil and Isle of Man, included a minimum sample size of approximately 1000. The datasets from the United States and especially Canada were substantially larger. Because of gender differences in abstention, actual *n* of male and especially of female current drinkers varied greatly and were small in some cases. Although women’s full samples were adequate (Table 1), there were small numbers of current female drinkers in India (37) and Sri Lanka (38) and, as noted, the latter had no heavy drinkers, precluding its inclusion in the women’s analyses. The response rates ranged from 38% to 96%, with a median of 64%; further details of the sampling design across countries is available in Wilsnack et al. [5]. Only current (past 12 months) drinkers were used in all of the analyses to follow, and the sample sizes available varied somewhat depending on the outcome under study.

### 2.2. Measures

Life-Area Harms and AUDIT Dependence Subscale: The two alcohol-related outcomes considered in these analyses are Life Area Harms [33,34] and AUDIT-based dependence [35,36], each of which is a sum of five 12-month items, where each sub-item was dichotomized to No (0) versus Yes (1). These two alcohol problem variables are based on the GENACIS Expanded Core questions. Identical or similar questions were asked in each participating country (except as noted). The surveys asked: “During the last 12 months, how often has your drinking had a harmful effect on your: work, studies or employment/marriage or family/friendships/physical health/and finances.” The five-item Life Areas Harmed (HARM-5) scale was the summative number of areas harmed with responses No/Yes: once or twice Yes and three or more times recoded to No vs. Yes (scale range 0 to 5; Cronbach’s α = 0.73 (men α = 0.74; women α = 0.67)), with alpha (a measure of internal reliability) being lower in each case where any item was removed. The AUDIT dependence items were termed by Graham et al. [37]: “Acute (but not highly endorsed) and chronic consequences experienced primarily by the drinker (labeled ‘personal’)”. The instructions prior to asking the AUDIT-5 items were: “During the last 12 months have you: (1) found you were not able to stop drinking once you had started, (2) failed to do what was normally expected of you because of your drinking, (3) needed a drink first thing in the morning to get yourself going after a heavy drinking session the night before, (4) experienced guilt or remorse after drinking, (5) been unable to remember what happened the night before because of your drinking?” The five-item AUDIT Dependence Scale was the summative number of items affirmed with the original responses Never/Less than once a month/Monthly/Daily or almost daily recoded to Never vs. Sometimes (range 0 to 5; Cronbach’s α = 0.71 (men α = 0.71; women α = 0.68), being lower in each case where any item was removed). For some countries, this recoded set (never vs. ever) of the responses was actually the form in which the response categories were asked [38].

Alcohol Consumption Measures (Volume, Binge Drinking, Usual Quantity, and Maximum Quantity): For each country included in the analysis, at least one of the two above problem variables were required as well as a measure of the overall yearly volume of consumption (recoded to number of 12 g drinks/year), frequency of binge drinking (defined below), and maximum quantity consumed in a single occasion. Volume was constructed across countries, typically using one of two sets of questions, and was based on recommendations proposed in the GENACIS dataset distribution. The first method was based on sums of beverage-specific volume measures [39], and the second was computed using generic quantity times frequency (QF) measures. The annual frequency of heavy episodic drinking (HED) or binge drinking was also created using differing schemes across countries, and the actual number of grams of ethanol per day defining a binge varied from around 50 to 70 g, with the majority of countries using 60 g to define a binge day. Binge frequency was not reported for any women from Sri Lanka, eliminating this country from the women’s multivariate analyses. The natural log of volume (i.e., ln(1 + volume)) was used as a predictor in place of volume. 

The maximum quantity consumed in a single occasion, obtained from the initial maximum question preceding a Graduated Frequencies-type measure [3], was typically asked in the following format: “Think of all kinds of alcoholic beverages combined, that is, any combination of cans, bottles or glasses of beer; glasses of wine; or drinks containing liquor of any kind (OR THE CULTURAL EQUIVALENT). During the last 12 months, what is the largest number of drinks you had on any single day? Was it: (7) 240 g or more of ethanol in a single day (20 or more drinks), (6) at least 144 g but less than 240 g (at least 12 but less than 20 drinks), (5) at least 96 g but less than 144 g (at least 8 but less than 12 drinks), (4) at least 60 g but less than 96 g (at least 5 but less than 8 drinks), (3) at least 36 g but less than 60 g (at least 3 but less than 5 drinks), (2) at least 12 g but less than 36 g (at least 1 but less than 3 drinks), (1) at least 1 g but less than 12 g (at least a sip but less than one full drink)”. This was the format of the item, for example, in Nigeria and in approximately 10 other countries. However, there were several permutations of this question format. Some countries did not include gram equivalents and only asked about the number of ‘drinks’ (e.g., Uganda, Mexico) with reference tables provided for drink size equivalents for the interviewer to assist the respondent. Other countries had slightly different ranges for the number of drinks for each category (e.g., Australia). Sweden had a single question “What is the largest number of drinks you drank on any one day?” and respondents provided a continuous number of drinks, and the question was only asked of a random half of the full sample. The US maximum item differed in that (a) it did not include the last category (i.e., a least a sip but less than a full drink) and (b) the drink ranges instead were about 1–2, 3–4, 5–7, 8–11, 12–23, and 24+ drinks/day instead of the ranges shown above [3,26]. Each of these variants in question formats as well as country-specific drink size recommendations in grams were harmonized and incorporated into the creation of a single maximum variable representing the midpoint of the drink range (or created range).

In the analyses, the frequency of binge drinking (HED) was compared to the maximum quantity, considered to be a measure of HID drinking patterns. A ‘continuous’ covariate with more possible response categories (i.e., less granularized) may outperform (in terms of variance explained) a similar covariate with fewer possible response categories simply due to its scale of measurement. Therefore, to make the two variables more equivalent in number of categories, we categorized HED binge frequency into seven binned values across countries defined as: (1) never, (2) 1–12 times per year, (3) 1–2 times per month, (4) 3–4 times per month, (5) 1–3 times per week, (6) every other day, and (7) every day.

A measure of usual quantity per drinking occasion was also constructed for each country using a frequency-weighted beverage-specific usual quantity. For each beverage consumed, the beverage frequency was multiplied by the beverage usual quantity. This product was summed across beverages and divided by the sum of the frequencies across beverages. This quantity could be computed for all countries except for Sweden, as beverage-specific and the maximum question were in different subsamples. The resulting usual quantity will be referred to subsequently as either usual quantity or a frequency-weighted beverage-specific usual quantity. This measure of usual quantity was constructed to be more consistent with maximum quantity than the single measure of usual quantity in a day, because the single usual quantity item was asked in varying formats across countries.

Individual Level Control Variables: Age and marital status were taken from responses to the GENACIS surveys in each country. Across countries, age was asked as a continuous variable. Marital status, although asked with slightly different possible categories across country, was coded as 1 if the respondent was married or living with a partner and 0 otherwise (mostly single but including some others). All analyses were performed disaggregated by gender.

### 2.3. Analysis

The primary goal of the present analyses is to examine whether the maximum quantity consumed in a day, considered to index HID, contributes to the prediction of alcohol-related problems across countries, above and beyond that of other typically used measures of overall intake (here log volume + 1) and HED (frequency of binge drinking). As only a relatively small number of countries were available, often 15 or fewer, multilevel models were not implemented due to the lack of available power for estimation of level 2 effects. In addition, due to the correlation of the various consumption measures, the focus was not primarily on the significance of the coefficient for the maximum but on the significance of the reduction in the model’s χ^2^ value as well as the increase in the model pseudo R^2^ estimate from including the maximum quantity consumed (i.e., improvement in model fit). The two primary outcomes under consideration were the five Life-Area Harms and the five Audit dependence items. Prior work [40] examined the functional form of the relationship between both overall ln volume of consumption as well as binge drinking and alcohol-related problems, using a non-parametric local smoothing algorithm, and found the relationships to be approximately linear. Therefore, ln volume and frequency of binge drinking were included as linear terms in the present analyses, with marital status and age as control variables in the gender disaggregated models.

For the AUDIT-5 and Harms-5 outcome variables, separate models were estimated for each country as well as overall across countries. For each outcome, a total of five models were estimated. First, a base model was estimated which controlled only for age and marital status, and the pseudo R^2^ estimate for the model was recorded. Second, ln volume was added as a covariate and the coefficient of ln volume and the increase in the model’s pseudo R^2^ estimate was noted. Third, frequency of binge drinking was included in the model along with age, marital status, and ln volume, and its coefficient and increase in the model pseudo R^2^ estimate was noted. Fourth, the maximum quantity consumed as the HID indicator was included in the model along with age, marital status, and ln volume, and its coefficient and increase in the model’s pseudo R^2^ estimate was noted. Finally, the maximum quantity consumed was included with age, marital status, ln volume, and frequency of binge drinking, and its coefficient and increase in the model’s pseudo R^2^ estimate was noted. The significance in each instance was determined by the reduction in the model’s χ^2^ value.

Considering the strong skewness in the raw outcome distribution for Life-Area Harms and AUDIT dependence items, each with response categories as integers ranging from 0 to 5, a Poisson regression model with a canonical log link function for the single parameter of the distribution (*λ*, serving as both the mean and variance of the distribution) was estimated for each country as well as overall across countries. This was appropriate as the outcome variable took on only non-negative integer values (0 to 5) with a strong concentration of values at 0 and with higher values represented with relatively low frequencies. Such a distribution closely resembles that of a Poisson distribution with a mean not far above zero. Although technically, a Poisson distribution has non-zero probability on all non-zero integers (i.e., 1, 2, 3, and on up to infinity) and our outcome mass function only has support for the integers from 0 to 5, we considered the departure of our empirical distribution from that of the theoretical Poisson, i.e., that the probability that a Poisson distribution takes on integer values of 6 or larger when the mean is as large as *λ* = 1 is only 0.0006. Even mean values as large as *λ* = 2 have less than 2% of their mass on integers above 5. From empirical plots of the Poisson probability mass function, overall mean parameter values were expected to be in the range of approximately 0.3. The preliminary results from base Poisson models indicated that the conditional mean values for those with higher problem levels could be as large as 1 but no larger than 2.

In a Poisson regression model, the natural log of the mean parameter (*λ*) is modeled as a function of individual level characteristics. For example, the specification for the final model including all covariates and using AUDIT-5 as the outcome variable was:$$ln(\lambda_{i,c})=\alpha_{c}+\theta_{1}\cdot A_{i,c}+\theta_{2}\cdot MS_{i,c}+\beta_{1}\cdot Ln(1+Volume_{i,c})+\beta_{2,c}\cdot Binge_{i,c}+\beta_{3,c}\cdot Max_{i,c}$$
where *λ_i_*_,*c*_ is the average of the Poisson distributed number of AUDIT items endorsed for the *ith* respondent in the *cth* country, *A_i_*_,*c*_ is their age, *MS_i_*_,*c*_ is their marital status, *Ln*(1 + *Volume_i_*_,*c*_) is their log volume of consumption, *Binge_i_*_,*c*_ is their frequency of binge drinking, and *Max_i_*_,*c*_ is their maximum quantity consumed in a day.

As the natural log of the mean of the Poisson distribution is modeled, interpretation of parameters must always be referenced to this fact. For example, within a given country *c*, the $\beta_{1,c}$ coefficient is the difference in log mean parameter values between two observations that differ in their log volume consumption score by 1 (note also that a difference of 1 between two logged values corresponds to larger or smaller differences in the untransformed volume or frequency of binge drinking, depending on the actual values of these untransformed variable). As the difference between two logged values is the log of the ratio, $\beta_{2,c}$ is also an estimate of the log ratio of the estimated mean of the outcome for a one-unit change in the Binge frequency scale.

## 3. Results

Table 1 shows the survey characteristics of countries with data on maximum quantities consumed, our HID indicator, and mean ages and percentage married or cohabiting, the two control variables. Table 2 and Table 3 show, for men and women separately, a number of characteristics of drinking and alcohol-related problem variables for each of the 17 countries in which data on maximum quantities were collected. Two countries did not ask all items in the AUDIT-5 scale (Mexico and Nicaragua) and three countries did not ask all items in the HARMS-5 scale (Finland, Isle of Man, and Mexico). As noted previously, no women reported any binge drinking in Sri Lanka and this subgroup was therefore excluded from analyses. The variables in Table 2 and Table 3 include the number and proportion of current (past 12 month) drinkers in the sample and averages among drinkers for each of maximum quantity, usual quantity per drinking day, frequency of binge drinking, and average volume per day. In addition, average scale values for the five-item AUDIT and HARM scales are also shown. Drinking rates varied dramatically across societies, with somewhat more variation for women than men. Average number of harms also varied dramatically, with men in Uruguay, Sweden, Canada, US, Australia, Brazil, and Argentina reporting the lowest averages and men in Uganda, Nicaragua, India, and Kazakhstan reporting the highest Average harms for women roughly followed those for men, with Argentinian women reporting the lowest average number of harms and Uganda the highest. For the AUDIT-5, the lowest averages were seen in Uruguay and the US, while the highest averages were found in Uganda, Kazakhstan, India, and Finland. For women, the lowest AUDIT-5 averages were seen in Argentina, Uruguay, and the US, while the highest average, by far, was for Ugandan women.

Note also that the interpretation of coefficients is with respect to changes in the *log* Poisson mean number of alcohol-related problems.

The distribution across countries of a categorized version of the maximum quantity, created for display purposes, is shown in Figure 1a for men and Figure 1b for women. The maximum categorized value of 1 (only present for a subset of countries) corresponds to maximum quantities of less than 1 drink; 2 roughly corresponds to 1–2 drinks; 3 corresponds to 3–4 drinks; 4 corresponds to 5–7 drinks; 5 corresponds to 8–11 drinks; 6 corresponds to 12–20/23 drinks; and 7 corresponds to 20/24+ drinks/day. The distributions show that the modal maxima for males showed large proportions of high intensity drinking at 12–20 drinks on any day for some countries (e.g., Kazakhstan, Nicaragua, Nigeria) and more evenly distributed or peaked at more moderate values for others, some having reasonable proportions in the 8–11 range (5 on the x-axis in the figures). The distribution of maximum quantity for women in Figure 1b tends to show lower modal maxima levels (for most countries) or was evenly distributed across the range.

Table 4 shows results for each country and overall for each of the five models predicting the AUDIT-5 scale for men. (Table 5 gives models predicting HARMS-5 for men.) For AUDIT-5, the base model (including only age and marital status as predictors), the model’s pseudo R^2^ ranged from quite low (0.001 to 0.01 in Nigeria, Sri Lanka, Uganda, Kazakhstan, and India) to somewhat higher (0.11 in Sweden and the USA).

For all countries, the beta coefficient for ln volume in the second model was highly significant, i.e., by reducing the model’s χ^2^ and increasing the pseudo R^2^ from 0.05 to 0.20. The results from Models 3 and 4 compare the relative improvement in the model fit from including HED frequency of binge drinking (Model 3) as opposed to instead adding the maximum quantity consumed (Model 4). For 8 of the 15 countries, the Model 3 results indicated that frequency of binge drinking was a significant predictor of the log Poisson mean AUDIT-5 score. Comparatively, the results from Model 4 indicated that using the maximum quantity consumed instead of frequency of binge drinking gave significant results for 11 of the 15 countries/societies, including 4 countries for which binge drinking was not significant (Costa Rica, Finland, Isle of Man, and Uruguay); the coefficients for all 4 countries were significant for the maximum quantity consumed (*p* < 0.001) in addition to 1 country for which binge was significant but the maximum quantity was not (India, *p* < 0.05 for binge). Including the maximum quantity consumed as a covariate above and beyond age, marital status, ln volume, *and* frequency of binge drinking (in Model 5) significantly improved the overall model fit for the same 11 countries for which the maximum alone had improved the model fit in Model 4 (Argentina, Australia, Canada, Costa Rica, Finland, Isle of Man, Kazakhstan, Nigeria, Sweden, Uruguay, and the USA). Note that although the direct interpretation of the coefficient for maximum may be problematic due to high correlations between volume, binge frequency, and maximum quantity consumed, a significant improvement to the model’s χ^2^ fit from including the maximum quantity consumed as a single variable covariate block indicates that significant additional information was added by the maximum measure.

Interpretation of the significant result for these 11 countries as well as for the 4 countries for which the maximum quantity consumed was not significant (Brazil, India, Sri Lanka, and Uganda) can be partially informed by considering the relationship between the maximum quantity and usual quantity of consumption. For male drinkers in each country (see Table 2) for each of the 11 countries for which significant improvements in model fitting were found by including the maximum quantity consumed as a predictor tended to have larger values of the average difference between the maximum and usual consumption, suggesting more high intensity drinking. In addition, the four countries for which the maximum quantity consumed did not improve the model fit tended to have lower average differences between the maximum and usual quantities across these countries. This may suggest, at least when predicting an alcohol dependence-type measure for men, countries for which there is high variance in drinking patterns may better diagnose alcohol problems by also collecting information on the maximum quantity (or episodes of HID) in addition to HED measures (i.e., frequency of 5+ binge drinking episodes).

In predicting the HARM-5 life-area harms for men, very similar results were found to those for predicting the AUDIT-5. Here, the five countries for which the maximum did not significantly improve the model fit were Brazil, India, Nicaragua, Sri Lanka, and Uganda. For example, in Table 2 the average difference between the maximum (13.7) and usual (11.0) quantities was also relatively small for Nicaragua compared to the other countries.

In predicting the AUDIT-5 scale for women (Table 6 below; Table 7 provides similar models for the HARM-5 outcome), the inclusion of the maximum quantity consumed as a predictor significantly improved the model fit for 12 of the 14 countries for which data were available (results not significant for India and Brazil).

Examining Table 3 (the analog of Table 2 but for women), India and Brazil again have average maxima and usual quantities consumed for women that are close to one another. For predicting the HARMS-5 scale for women, slightly different results were found, with only five countries being significant for Model 5 (testing the addition of Maximum after accounting for HED). Here, the maximum quantity consumed did not significantly improve the model fit for seven countries including Brazil, India, Kazakhstan, Nicaragua, Nigeria, Sweden, and the US (with results marginally significant for Uganda). For at least two of these countries, Kazakhstan and Nigeria, the difference between the maximum and usual quantities was relatively large in comparison to other countries where the maximum quantity was a significant predictor, suggesting a less important role of the maximum quantity for women in relation to life-area harms. Still, among women in Model 4, when the maximum quantity was added to volume and the base model (omitting HED), seven countries showed incremental significance for the high intensity drinking measure, the added countries being Kazakhstan and the US. Recall that in Sweden, the usual quantity per drinking occasion and maximum were asked in differing subsamples, so the comparison of their values was not useful.

## 4. Discussion

Considering the results by gender, each of the respective societies for which significant improvements in model fit were found by including the maximum quantity consumed as a predictor tended to have larger differences between the maximum and usual consumption. This implies considerable variation in the amounts consumed on any given day, including high intensity drinking above 5+ drinks (8–11, 12+, and in some societies an even higher number of drinks). This remains true when controlling for HED (here defined as 5+ drinks/day).

Thus, among current drinkers in many countries, both in the developed and developing world, the maximum on any day, especially when including HID levels such as 8–11 and 12+ drinks, is a sensitive measure that adds to the prediction of alcohol dependence and tangible harms in the lives of drinkers and their families. For a substantial number of people across the world, the quantities consumed at least sometimes exceeded the hazardous level of 5 or more drinks, indicating patterns with high intensity drinking. Assessing the maximum quantity consumed on any given day in the prior year is relatively easy [26], and is indicated for monitoring and surveillance programs [6]. By capturing unusually large amounts (generally indicative of high variability in drinking), the maximum quantity augments the mean volume and the 5+ binge measures that have been almost universally adopted as pattern indicators in recent years (or 5+/4+ for men/women as often seen in US studies, following Wechsler et al. [41]). We recommend, as did the Taskforce on Recommended Questions for the US of NIAAA [42], that surveys in other countries include the maximum quantity consumed in the minimal set of alcohol items in monitoring and epidemiological surveys. Along with mean volume, the maximum—even without assessing 5+ binge drinking—performs very well as a pattern measure, potentially sharpening the predictions of alcohol-related harms [2]. In this study, this is the case for men and women regarding Alcohol Use Disorder symptoms, but more for men than women regarding associations with Life-Area Harms. The basis for the gender difference in this case may reflect the fact that women generally have lower maxima than men, but the gender difference seen for Life-Area Harms deserves more study. Clearly, however, the overall policy implications are to design alcohol control measures with the capacity to bring down both the number of heavy days (consuming five or more drinks) and the peak drinking level (i.e., to reduce high intensity drinking). We know that population measures affecting overall drinking may pull down heaviest drinkers too [43]. However, the question remains whether additional interventions to reduce HID specifically may also be needed, such as normative approaches [44,45], work interventions [46], SBIRT [47], or other targeted strategies like mandating server intervention training [48] and social host liability [49].

Recently, a US study found that quality screening, defined as health care practitioners asking about quantity of alcohol consumed, was more likely for those with HID rather than just HED [50], This is a promising finding that suggests that the practitioners were at least somewhat sensitive to health problems stemming from high intensity drinking. Reducing HID (as well as HID) is an important goal of treatment for alcohol use disorders to reduce likelihood of relapse [27,29].

Among the limitations of this study are its reliance on self-reported cross-sectional data and a limited set of countries with available data for analysis. In addition, the surveys were conducted at varying times from 1998 (Mexico) to 2007 (Australia)—mostly between 2000 and 2003. The surveys, though based on a common instrument, used differing modalities of data collection (see Table 1). The frames of the surveys varied from nationally representative adult population surveys to subnational area surveys. Although having a few differences in item formats and some cultural adaptations as needed, the GENACIS questionnaires were highly similar by design, and were translated/back translated with a common protocol. For this study we harmonized the variables used for analysis. A potential limitation is that while age and marital status, both important covariates, were controlled, and the models were disaggregated by gender, a wider range of demographic variables might also be influential. A strength of this study is that this appears to be the first investigation of the potential association between alcohol-related problems and the maximum measure, taken as an indication of high intensity drinking, considering both men and women separately, and taking account of more conventional drinking pattern variables and two key demographics, in a range of societies with varying income levels.

## 5. Conclusions

After accounting for age, marital status, binge drinking (5+) rate, and average alcohol intake, across a broad range of high and low income societies located in multiple continents, the results support the importance of measures of high intensity drinking (far beyond 5+) for enhancing associations to both alcohol dependence symptoms and negative effects on health, marital, or friendship relationships, work or study, and finances. In 75–85 percent of the included countries, high intensity drinking was incrementally significant in relation to dependence symptoms (more so for women than men). In relation to life areas harmed, from about two thirds to two fifths of the included countries (favoring men) showed an incremental relationship owing to high intensity drinking. These gender differences, especially for Life-Area Harms, deserve further study. Given the broad findings of the advantage of adding an indicator of high intensity drinking in models predicting alcohol-related problems, we recommend adding the maximum quantity consumed on any day to the more typical average volume and 5+ binge drinking measures in future multinational epidemiological surveys of alcohol use patterns and related harms. More fully capturing variation in drinking amounts over time, as indicated by high intensity drinking for any given volume of consumption, is vital for understanding how both chronic and acute alcohol problems accrue. As we have shown, in many parts of the world, alcohol dependence, health harms, and hurtful effects on people and families are all exacerbated by the scourge of high intensity drinking. Our results support policy makers in adopting both evidence-informed population-wide alcohol control measures and more targeted interventions aimed at curbing extremely heavy drinking levels, shown here to be widely linked to alcohol dependence symptoms and other medical and social harms.

## Figures and Tables

**Figure 1 ijerph-20-03748-f001:**
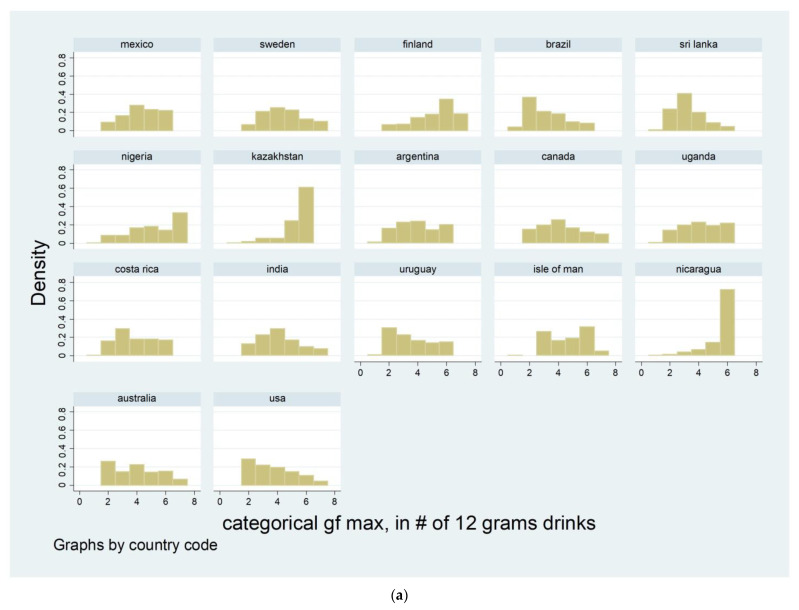
(**a**) Maximum Quantity Distribution among male drinkers. (Note: see x-axis key under female drinkers charts; y-axis is proportion). (**b**) Maximum Quantity Distribution among female drinkers. **X-Axis:** Sub-HID levels: 1 = <1 drink, 2 = 1–2 drinks, 3 = 3–4 drinks, 4 = 5–7 drinks; HID levels: 5 = 8–11 drinks, 6 = 12–20* drinks, 7 = 20+* drinks (*12–24 and 24+ drinks for Sweden and the USA).

**Table 1 ijerph-20-03748-t001:** Survey Sample Characteristics.

Country	Survey Year	Total Women (n)	Total Men (n)	Mean Age (Years)	Percent Married/ Cohabit.	Sampling Frame	Survey Mode
Argentina	2003	598	401	39.9	61	Regional: ≈95% of population (Buenos Aires City and province)	Face-to-face
Australia	2007	1221	831	43.9	61	Regional: (Victoria)	Telephone
Brazil	2001/ 2002	387	273	37.7	71	Regional: (Botucatu, Sao Paulo State)	Face-to-face
Canada	2004	6904	5360	43.2	60	National	Telephone
Costa Rica	2003	776	381	37.1	58	Regional: ≈50% of population (Greater Metropolitan Area)	Face-to-face
Finland	2000	987	945	42.9	66	National	Face to face
India	2003	1215	1318	32.2	70	Regional: (Karnataka, five regions including Bangalore)	Face-to-face
Isle of Man	2006	425	366	46.2	72	National	Mixed mode (57.5% F-to-F; 42.5% Tel)
Kazakhstan	2002/2003	545	487	41.5	70	Regional (East Kazakhstan)	Face-to-face
Mexico	1998	3329	2382	57.4	67	National	Face-to-face
Nicaragua	2005	1390	594	34.5	61	Regional: (Bluefields, Esteli, Juigalpa, Leon, and Rivas)	Face-to-face
Nigeria	2003	926	1068	37.4	73	Regional: (two South, three North states and Federal Capital)	Face-to-face
Sri Lanka	2002	552	543	39.8	73	Near National: (17 of 25 districts)	Face-to-face
Sweden	2002	954	870	41.2	65	National	Telephone
Uganda	2003	743	695	32.6	57	Regional: (one district in each of four regions)	Face-to-face
Uruguay	2004	624	376	40.6	57	National	Face-to-face
USA	2000	3338	3057	39.7	65	National	Telephone

**Table 2 ijerph-20-03748-t002:** Male Drinking and Problem Estimates.

Country/Survey	N of Drinkers	% Current Drinkers	Average Maximum Quantity ^a^	Average Usual Quantity ^a^	Average Freq Binge ^b^	Average Volume/Day ^c^	Average AUDIT-5	Average HARMS-5
Argentina	368	91.5	7.91	2.86	30.28	1.28	0.42	0.30
Australia	882	88.2	7.60	2.82	16.36	0.98	0.59	0.26
Brazil	325	58.2	5.14	3.78	44.67	1.73	0.43	0.28
Canada	4855	81.7	8.33	2.95	24.58	1.34	0.62	0.17
Costa Rica	285	68.5	6.80	4.43	15.43	0.90	0.74	0.61
Finland	871	92.2	9.47	3.68	17.54	1.07	1.25	--
India	492	36.7	5.75	3.71	99.73	2.91	1.26	0.98
Isle of Man	421	92.9	9.70	4.48	10.01	2.08	0.47	--
Kazakhstan	405	75.1	11.58	5.72	33.07	1.70	1.30	0.91
Mexico	1833	77.0	9.19	3.60	36.19	1.18	--	--
Nicaragua	266	43.3	13.71	11.03	49.68	1.83	--	1.41
Nigeria	467	42.1	13.39	4.47	75.25	2.65	0.64	0.60
Sri Lanka	323	53.6	4.69	4.33	15.19	1.82	0.44	0.42
Sweden	762	88.7	7.88	--	12.83	0.57	0.52	0.16
Uganda	393	54.6	7.12	5.00	63.30	3.61	1.64	1.67
Uruguay	305	81.1	8.48	3.08	25.26	1.53	0.26	0.15
USA	2307	67.0	7.91	2.75	24.94	1.20	0.26	0.19
All Countries	15,560	71.7	8.41	3.53	29.73	1.41	0.64	0.36

^a^ In 12 g ethanol drinks, ^b^ in days/year, ^c^ drinks/day (12-month average), -- data unavailable.

**Table 3 ijerph-20-03748-t003:** Female Drinking and Problem Estimates.

Country/Survey	N of Drinkers	% Current Drinkers	Average Maximum Quantity ^a^	Average Usual Quantity ^a^	Average Freq Binge ^b^	Average Volume/Day ^c^	Average AUDIT-5	Average HARMS-5
Argentina	441	73.7	3.28	1.26	0.92	0.25	0.09	0.03
Australia	1172	81.7	4.10	1.95	5.64	0.49	0.49	0.29
Brazil	283	41.3	4.37	2.75	34.57	1.23	0.26	0.11
Canada	5891	74.8	4.56	1.29	7.27	0.57	0.37	0.09
Costa Rica	367	42.8	4.21	2.66	4.32	0.27	0.35	0.21
Finland	889	90.2	5.11	2.18	5.81	0.36	0.62	--
India	37	3.0	1.85	1.59	52.53	1.32	0.30	0.30
Isle of Man	471	86.3	6.57	2.81	2.71	0.81	0.29	--
Kazakhstan	402	63.7	6.26	2.43	7.61	0.23	0.36	0.28
Mexico	1406	42.2	3.59	2.36	3.62	0.15	--	--
Nicaragua	149	10.5	8.79	6.11	18.65	0.94	--	0.60
Nigeria	213	22.3	11.63	3.41	60.57	2.25	0.54	0.62
Sri Lanka	38	6.4	1.38	1.39	0.0	0.08	0.03	0.14
Sweden	748	80.7	4.07	--	4.47	0.30	0.28	0.08
Uganda	301	39.7	3.99	3.50	20.87	1.11	1.08	0.83
Uruguay	376	60.3	4.46	1.68	3.39	0.43	0.10	0.04
USA	2276	56.2	4.41	2.09	6.47	0.39	0.14	0.10
All Countries	15,460	56.2	4.59	2.16	7.67	0.50	0.34	0.14

^a^ in 12 g ethanol drinks, ^b^ in days/year, ^c^ drinks/day (12-month average), -- data unavailable.

**Table 4 ijerph-20-03748-t004:** Male Regressions Predicting AUDIT-5.

Outcome: AUDIT-5	Base Model ^a^	Base + Ln Volume		Base + Ln Volume + Binge		Base + Ln Volume + Max		Base + Ln Volume + Binge + Max	
Country/Survey	R^2^	Beta ^b^	ΔR^2^	Beta ^c^	ΔR^2^	Beta ^d^	ΔR^2^	Beta ^e^	ΔR^2^
Argentina	0.06	0.57 ***	0.13	0.004 ***	0.03	0.16 ***	0.08	0.25 ***	0.16
Australia	0.03	0.49 ***	0.06	0.003 ***	0.02	0.09 ***	0.05	0.08 ***	0.04
Brazil	0.02	0.46 ***	0.14	0.001	0.003	0.001	0.01	−0.17	0.01
Canada	0.07	0.54 ***	0.10	0.002 ***	0.003	0.12 ***	0.04	0.11 ***	0.04
Costa Rica	0.03	0.50 ***	0.18	−0.002	0.001	0.12 ***	0.06	0.12 ***	0.06
Finland	0.03	0.42 ***	0.10	0.003	0.001	0.09 ***	0.04	0.09 ***	0.04
India	0.006	0.49 ***	0.20	0.001 *	0.002	−0.004	0.001	−0.005	0.000
Isle of Man	0.08	0.63 ***	0.12	0.001	0.001	0.07 ***	0.02	0.07 ***	0.03
Kazakhstan	0.005	0.31 ***	0.10	0.002 ***	0.01	0.12 ***	0.05	0.12 ***	0.05
Mexico	--	--	--	--	--	--	--	--	--
Nicaragua	--	--	--	--	--	--	--	--	--
Nigeria	0.001	0.48 ***	0.11	0.002 ***	0.01	0.06 **	0.04	0.05 **	0.03
Sri Lanka	0.002	0.24 ***	0.05	0.001	0.01	0.008	0.001	0.001	0.000
Sweden	0.11	0.55 ***	0.09	0.005 ***	0.01	0.11 ***	0.03	0.10 ***	0.03
Uganda	0.003	0.29 ***	0.08	−0.001	0.001	−0.01	0.003	−0.01	0.001
Uruguay	0.05	0.50 ***	0.10	0.002	0.003	0.07 ***	0.03	0.07 ***	0.003
USA	0.11	0.54 ***	0.13	0.004 ***	0.03	0.04 ***	0.03	0.03 ***	0.01
All Countries	0.04	0.45 ***	0.12	0.002 ***	0.01	0.04 ***	0.02	0.04 ***	0.01

^a^ Base model controls for age and marital status. Note: * *p* < 0.05, ** *p* < 0.01, *** *p* < 0.001 (based on reduction in model χ^2^). ^b^ Beta is for the Ln Volume term where model includes age and marital status. ^c^ Beta is for the Binge term where model includes age, marital status, and ln volume. ^d^ Beta is for the GF Max term where model includes age, marital status, and ln volume. ^e^ Beta is for the GF Max term where model includes age, marital status, ln volume, and binge.

**Table 5 ijerph-20-03748-t005:** Male Regressions Predicting HARMS-5.

Outcome: HARMS-5	Base Model ^a^	Base + Ln Volume		Base + Ln Volume + Binge		Base + Ln Volume + Max		Base + Ln Volume + Binge + Max	
Country/Survey	R^2^	Beta ^b^	ΔR^2^	Beta ^c^	ΔR^2^	Beta ^d^	ΔR^2^	Beta ^e^	ΔR^2^
Argentina	0.03	0.44 ***	0.07	0.004 ***	0.03	0.16 ***	0.08	0.28 ***	0.18
Australia	0.08	0.36 ***	0.03	0.005 ***	0.04	0.07 ***	0.02	0.04 *	0.01
Brazil	0.04	0.65 ***	0.19	0.001	0.000	0.02	0.001	0.008	0.001
Canada	0.06	0.68 ***	0.10	0.004 ***	0.02	0.15 ***	0.04	0.13 ***	0.03
Costa Rica	0.05	0.48 ***	0.14	0.001	0.001	0.11 ***	0.04	0.11 ***	0.04
Finland	--	--	--	--	--	--	--	--	--
India	0.02	0.49 ***	0.17	0.001	0.000	0.01	0.001	0.01	0.001
Isle of Man	--	--	--	--	--	--	--	--	--
Kazakhstan	0.01	0.29 ***	0.08	0.002 **	0.01	0.06 ***	0.02	0.06 ***	0.02
Mexico	--	--	--	--	--	--	--	--	--
Nicaragua	0.001	0.18 ***	0.03	−0.001	0.001	0.01	0.001	0.01	0.004
Nigeria	0.01	0.56 ***	0.14	0.001 *	0.01	0.06*	0.03	0.05 *	0.02
Sri Lanka	0.002	0.45 ***	0.18	0.001	0.000	0.03	0.01	0.03	0.003
Sweden	0.18	0.55 ***	0.05	0.003 *	0.005	0.15 ***	0.04	0.15 ***	0.04
Uganda	0.01	0.27 ***	0.07	0.001	0.001	0.002	0.001	−0.004	0.001
Uruguay	0.02	0.84 ***	0.19	−0.001	0.000	0.12 ***	0.06	0.13 ***	0.07
USA	0.08	0.45 ***	0.09	0.004 ***	0.02	0.04 ***	0.02	0.02 **	0.02
All Countries	0.03	0.54 ***	0.13	0.002 ***	0.01	0.02 ***	0.02	0.02 ***	0.03

^a^ Base model controls for age and marital status. Note: * *p* < 0.05, ** *p* < 0.01, *** *p* < 0.001 (based on reduction in model χ^2^). ^b^ Beta is for the Ln Volume term where model includes age and marital status. ^c^ Beta is for the Binge term where model includes age, marital status, and ln volume. ^d^ Beta is for the GF Max term where model includes age, marital status, and ln volume. ^e^ Beta is for the GF Max term where model includes age, marital status, ln volume, and binge.

**Table 6 ijerph-20-03748-t006:** Female Regressions Predicting AUDIT-5.

Outcome: AUDIT-5	Base Model ^a^	Base + Ln Volume		Base + Ln Volume + Binge		Base + Ln Volume + Max		Base + Ln Volume + Binge + Max	
Country/Survey	R^2^	Beta ^b^	ΔR^2^	Beta ^c^	ΔR^2^	Beta ^d^	ΔR^2^	Beta ^e^	ΔR^2^
Argentina	0.14	0.52 ***	0.10	0.05 ***	0.07	0.21 ***	0.17	0.19 ***	0.13
Australia	0.03	0.21 ***	0.03	0.004 ***	0.002	0.12 ***	0.07	0.09 ***	0.03
Brazil	0.04	0.72 ***	0.23	−0.001	0.003	−0.02	0.001	−0.003	0.000
Canada	0.09	0.62 ***	0.10	0.003 ***	0.004	0.11 ***	0.02	0.10 ***	0.02
Costa Rica	0.06	0.33 ***	0.06	0.004	0.001	0.10 ***	0.03	0.11 ***	0.03
Finland	0.09	0.55 ***	0.13	0.003 *	0.003	0.12 ***	0.04	0.12 ***	0.004
India	0.04	0.75 ***	0.30	0.005 ^†^	0.06	−0.54	0.01	−0.49	0.01
Isle of Man	0.11	0.57 ***	0.09	0.007	0.003	0.10 ***	0.05	0.14 ***	0.07
Kazakhstan	0.01	0.44 ***	0.12	0.004 ***	0.02	0.09 ***	0.04	0.08 ***	0.03
Mexico	--	--	--	--	--	--	--	--	--
Nicaragua	--	--	--	--	--	--	--	--	--
Nigeria	0.03	0.64 ***	0.21	0.001 ^†^	0.006	0.04 *	0.02	0.04 *	0.02
Sri Lanka	--	--	--	--	--	--	--	--	--
Sweden	0.15	0.47 ***	0.06	0.003	0.001	0.07 **	0.02	0.07 **	0.01
Uganda	0.01	0.23 ***	0.07	0.002 *	0.01	0.02 ^†^	0.02	0.03 *	0.01
Uruguay	0.05	0.66 ***	0.14	0.01 ***	0.08	0.13 ***	0.08	0.10 ***	0.03
USA	0.10	0.57 ***	0.14	0.002 **	0.01	0.06 ***	0.04	0.06 ***	0.03
All Countries	0.06	0.49 ***	0.11	0.003 ***	0.01	0.06 ***	0.02	0.06 ***	0.01

^a^ Base model controls for age and marital status. Note: ^†^ *p* < 0.1, * *p* < 0.05, ** *p* < 0.01, *** *p* < 0.001 (based on reduction in model χ^2^). ^b^ Beta is for the Ln Volume term where model includes age and marital status, ^c^ Beta is for the Binge term where model includes age, marital status, and ln volume. ^d^ Beta is for the GF Max term where model includes age, marital status, and ln volume. ^e^ Beta is for the GF Max term where model includes age, marital status, ln volume, and binge.

**Table 7 ijerph-20-03748-t007:** Female Regressions Predicting HARMS-5.

Outcome: HARMS-5	Base Model ^a^	Base + Ln Volume		Base + Ln Volume + Binge		Base + Ln Volume + Max		Base + Ln Volume + Binge + Max	
Country/Survey	R^2^	Beta ^b^	ΔR^2^	Beta ^c^	ΔR^2^	Beta ^d^	ΔR^2^	Beta ^e^	ΔR^2^
Argentina	0.03	0.60 ***	0.13	0.02	0.01	0.15 **	0.05	0.15 *	0.04
Australia	0.07	0.30 ***	0.04	0.004 *	0.01	0.10 ***	0.03	0.08 ***	0.02
Brazil	0.10	0.74 ***	0.16	−0.003	0.001	−0.12	0.01	−0.08	0.002
Canada	0.09	0.86 ***	0.11	0.004 ***	0.02	0.15 ***	0.03	0.13 ***	0.02
Costa Rica	0.07	0.37 ***	0.07	0.001	0.000	0.18 ***	0.10	0.19 ***	0.10
Finland	--	--	--	--	--	--	--	--	--
India	0.07	0.42 ***	0.15	0.003	0.002	−0.37	0.005	−0.34	0.02
Isle of Man	--	--	--	--	--			--	--
Kazakhstan	0.01	0.37 ***	0.09	0.004 ***	0.02	0.04 *	0.02	0.02	0.002
Mexico	--	--	--	--	--	--	--	--	--
Nicaragua	0.002	0.23 ***	0.03	−0.008 *	0.03	−0.02	0.002	−0.01	0.001
Nigeria	0.06	0.44 ***	0.12	0.001	0.001	−0.01	0.002	−0.02	0.003
Sri Lanka	--	--	--	--	--	--	--	--	--
Sweden	0.08	0.42 **	0.03	−0.02	0.001	0.03	0.001	0.04	0.002
Uganda	0.01	0.16 ***	0.03	−0.001	0.002	0.02	0.004	0.03 ^†^	0.02
Uruguay	0.06	0.81 ***	0.15	0.004	0.003	0.14 ***	0.09	0.15 ***	0.08
USA	0.09	0.64 ***	0.14	0.003 ***	0.02	0.03 *	0.02	0.01	0.003
All Countries	0.05	0.54 ***	0.10	0.004 ***	0.02	0.05 ***	0.02	0.04 ***	0.02

^a^ Base model controls for age and marital status. Note: ^†^ *p* < 0.1, * *p* < 0.05, ** *p* <0.01, ****p* < 0.001 (based on reduction in model χ^2^). ^b^ Beta is for the Ln Volume term where model includes age and marital status. ^c^ Beta is for the Binge term where model includes age, marital status, and ln volume. ^d^ Beta is for the GF Max term where model includes age, marital status, and ln volume. ^e^ Beta is for the GF Max term where model includes age, marital status, ln volume, and binge.

## Data Availability

GENAHTO data are available to researchers under data sharing agreements and subject to certain country permissions: https://genahto.org/contact-us/ (accessed on 28 December 2022).

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
