# Peer review of "High Intensity Drinking (HID) Assessed by Maximum Quantity Consumed Is an Important Pattern Measure Adding Predictive Value in Higher and Lower Income Societies for Modeling Alcohol-Related Problems"

_ijerph, 2023, doi:10.3390/ijerph20043748_

Round 1
Reviewer 1 Report
This study used data from a multicenter database and found HID is a valuable predictor of problematic alcohol use, based on two 2 main results: 1) adding HID improved model fit (table 2-5); 2) higher HID suggests greater drinking variability (figure 2a-b). Based on these findings, authors suggest to add HID into the alcohol use disorder screening tool, and HID can also be used as a treatment target. This manuscript has the following issues:
Major issues:
1. No hypothesis. Authors already pointed out HID has been used as a predictor of AUD (lines 53-63), then what are the purposes and hypotheses for this current study are unclear.
2. It is not clear how HID is different from (or better than) HED or binge drinking.
3. The author built 5 models and compared models based on R2 changes, however, did not clarify how to decide whether the ΔR2 change is significant, or which model is better; in addition, it is not clear how the ΔR2 was calculated in tables 2-4 (all compared to the base model)?
4. According to figure 2a-b and its interpretation, seems that HID suggests greater drinking variability, if that is the case, why use HID but not the variability (e.g., Max - Usual Q) as the predictor?
Minor issues:
1. Didn’t spell full name for HED, AUDIT, GENACIS in first place.
2. The background of GENACIS is not clear.
3. Demographics of the participants are missing, e.g., age, income level, education, sex, etc.
4. Writing is overall wordy, consider using more tables/graphs to replace text.
5. Authors claimed several key demographics were taken into account (line 476), but it is not clear what are the “key demographics” other than sex and country.
Author Response
Below each Reviewer comment we have provided our responses.
Reviewer 1
Comments and Suggestions for Authors
This study used data from a multicenter database and found HID is a valuable predictor of problematic alcohol use, based on two 2 main results: 1) adding HID improved model fit (table 2-5); 2) higher HID suggests greater drinking variability (figure 2a-b). Based on these findings, authors suggest to add HID into the alcohol use disorder screening tool, and HID can also be used as a treatment target. This manuscript has the following issues:
Major issues:
- No hypothesis. Authors already pointed out HID has been used as a predictor of AUD (lines 53-63), then what are the purposes and hypotheses for this current study are unclear.
RESPONSE: We agree and have added a rationale for the study (its purpose beyond earlier work) and followed this with a statement of the hypothesis, both added at the end of the Introduction, as follows:
“However, there is a dearth of multinational studies on HID, and especially on its relationship to alcohol-related problems. Based on prior studies limited to the US [27] and its southern border [28], we hypothesized that HID, here indexed by maximum amount drunk, would significantly augment standard intake and binge (5+) drinking measures, improving the prediction of symptoms of AUD and life-area harms in a wide range of low, middle and high income countries.’
- It is not clear how HID is different from (or better than) HED or binge drinking.
RESPONSE: Earlier in the Introduction we have made several edits to sharpen our explanation of why HID, at amounts going well beyond the HED 5+ level, are important for predicting alcohol problems (see edits in Lines 36-39). In addition, the empirical study presented here specifically aims to test whether this assertion, that adding HID to the more standard HED threshold, does in fact add significantly to the prediction of two such problem measures assessing alcohol dependence symptoms and life area harms.
- The author built 5 models and compared models based on R2 changes, however, did not clarify how to decide whether the ΔR2 change is significant, or which model is better; in addition, it is not clear how the ΔR2 was calculated in tables 2-4 (all compared to the base model)?
RESPONSE: Thanks for identifying a potential for unclarity. Although covered in Methods-Analysis, where we noted that the significance test (in each case) was based on a reduction (improvement) in Chi-square model fit) we decided to emphasize the nature of the test for significance of the pseudo R-square change from (for example) adding Maximum to the full Model 4 set of independent variables (i.e., Model 5), by adding the basis of the significance test also to Results as well as, importantly, in the footnotes to Tables 4, 5, 6, & 7 where significance levels are identified by adding after these a paren: “(based on reduction in model χ2). We realize many readers might look mainly at tables and selected parts of a paper, and so this redundancy makes it more likely that the basis of the significance tests in each case will be seen.
- According to figure 2a-b and its interpretation, seems that HID suggests greater drinking variability, if that is the case, why use HID but not the variability (e.g., Max - Usual Q) as the predictor?
RESPONSE: This is an interesting observation and question. Our main point in the study was to empirically demonstrate the utility, in predicting two types of alcohol problems (dependence symptoms and life area harms scales), of high intensity drinking assessed by maximum over and above standard measures (ln volume and 5+). Having found this (especially for males) across many of the included societies, we sought to offer a potential reason that for certain countries maximum played less of an incremental role (i.e., did not add significantly to model fit). One basis we observed was when usual quantity and maximum were similar, maximum might play less of a role. In our revised manuscript we realized that adding figures 2a and 2b placed too much emphasis on this potential explanation. Since the data (average quantity and average maximum) are given for men in Table 2 and for women in Table 3 we eliminated (also for parsimony) the Figures 2a and 2b and instead pointed the reader to Tables 2 and 3. In addition to make this point we now identify in Results the male values of usual quantity and maximum for Nicaragua (Lines 326-327): “For example, in Table 2 the average difference between maximum (13.7) and usual (11.0) quantities is also relatively small for Nicaragua compared to the other countries.”
Minor issues:
- Didn’t spell full name for HED, AUDIT, GENACIS in first place.
RESPONSE: Thank you. We have now done so.
- The background of GENACIS is not clear.
RESPONSE: at the beginning of Methods Data Sources GENACIS has been clarifies, also providing references and website:
“Survey samples come from those collected under the multi-country GENACIS (Gender, Alcohol and Culture, an International Study) project [5, 38-40]. Questions were developed first in English, then translated into the main language of the society surveyed, followed by back-translation with a check for accuracy and culturally accessible meaning. Guidelines for question translation were adapted from World Health Orgaization (WHO) strategies (see the GENACIS website: https://med.und.edu/genacis/ and Wilsnack et al. [5]).”
Table 1 is also useful in characterizing GENACIS datasets used here.
- Demographics of the participants are missing, e.g., age, income level, education, sex, etc.
RESPONSE: As we clarify, two demographic covariates (besides sex, which was disaggregated in the models) were included—age and Marital Status (a dummy for married/cohabiting vs other status (mostly single). At the reviewer’s suggestion we have added sample mean age and mean prevalence of Married/cohabiting to Table 1.
- Writing is overall wordy, consider using more tables/graphs to replace text.
RESPONSE: We agree and felt Methods was most wordy so we edited this section to reduce wordiness and to eliminate some un-needed text. However, we did not see a way to convert text to more tables and graphs. Still, the overall paper length has also been reduced by eliminating redundant Figured 2a and 2b.
- Authors claimed several key demographics were taken into account (line 476), but it is not clear what are the “key demographics” other than sex and country.
RESPONSE: Thank you for pointing out that ‘several’ was imprecise. We now make clear that the demographic controls in the separate models by sex are two—age and Marital Status (Married/Cohabiting vs Other). We now also note that a limitation is that other demographic variables might also have been influential.
Reviewer 2 Report
the manuscript by Thomas K Greenfield and colleagues reports interesting research on the advantages of including an indicator of heavy drinking (the maximum) to describe the drinking pattern, in models predicting alcohol-related problems. The results support the importance of measures of high-intensity drinking (far beyond 5+) for enhancing associations to both alcohol dependence symptoms and negative effects on health and society.
The manuscript is well-written, the background is appropriate, the methods are clearly described, and the results are sound.
The discussion is very short, and I think it would benefit from a more broad consideration of the effects of heavy alcohol drinking, from both a preclinical and clinical point of view.
Author Response
Reviewer 2
We have responded to each point below
Comments and Suggestions for Authors
the manuscript by Thomas K Greenfield and colleagues reports interesting research on the advantages of including an indicator of heavy drinking (the maximum) to describe the drinking pattern, in models predicting alcohol-related problems. The results support the importance of measures of high-intensity drinking (far beyond 5+) for enhancing associations to both alcohol dependence symptoms and negative effects on health and society.
The manuscript is well-written, the background is appropriate, the methods are clearly described, and the results are sound.
RESPONSE: Thank you for the endorsement.
The discussion is very short, and I think it would benefit from a more broad consideration of the effects of heavy alcohol drinking, from both a preclinical and clinical point of view.
RESPONSE: Because of the Detail required for presenting results which include some post hoc explanations, we chose to keep the Discussion succinct. However, we take the point regarding clinical and preclinical significance and have added the following text to the discussion, adding reference [54]:
“Recently, a US study found that quality screening, defined as health care practitioners asking about quantity of alcohol consumed, was more likely for those with HID, rather than just HED [54], This is a promising finding that suggests that the practitioners were at least somewhat sensitive to health problems stemming from high intensity drinking. Reducing HID (as well as HID) is an important goal of treatment for alcohol use disorder to reduce likelihood of relapse [27, 29].”
54 Subbaraman, M.S., C.K. Lui, K.J. Karriker-Jaffe, T.K. Greenfield, and N. Mulia, Predictors of alcohol screening quality in a US general population sample and subgroups of heavy drinkers. Preventive Medicine Reports, 2022, 29: 101932.
Round 2
Reviewer 1 Report
I do not have more comments at this time.